# Seismic Performance Analysis of Segmental Assembled Concrete-Filled Steel Tubular Pier with External Replaceable Energy Dissipation Ring

**Chengquan Wang** [1,2,3], **Zheng Qu** [4], **Yun Zou** [4,*], **Chongli Yin** [4], **Yanwei Zong** [4] **and Zexuan Sun** [4]

1. Department of Civil Engineering, Zhejiang University City College, Hangzhou 310015, China; wcqdesign@163.com
2. Zhejiang Engineering Research Center of Intelligent Urban Infrastructure, Hangzhou 310015, China
3. Key Laboratory of Safe Construction and Intelligent Maintenance for Urban Shield Tunnels of Zhejiang Province, Hangzhou 310015, China
4. School of Environment and Civil Engineering, Jiangnan University, Wuxi 214122, China; q15261516958@163.com (Z.Q.); y15251636986@163.com (C.Y.); zyw320830@163.com (Y.Z.); szx720318@163.com (Z.S.)
* Correspondence: zouyun_22@126.com

**Abstract:** In order to develop a new type of prefabricated bridge structure system with green, efficient and recoverable function, which complies with the new requirements of rapid repair of pier function after earthquake, and improves the applicable performance of prefabricated assembled pier in medium and high-intensity seismic areas, a precast segmental concrete-filled steel tubular (PSCFST) pier with an external energy dissipation ring is proposed. Based on ABAQUS analysis software, a four-segment PSCFST pier model is established, and the pseudo-static comparative analysis is carried out between the traditional PSCFST pier and the PSCFST pier with the external energy dissipation ring. The results show that compared with the traditional PSCFST pier, the lateral bearing capacity of the PSCFST pier with an external energy dissipation ring is increased by 60%, the energy dissipation capacity is increased by about 20 times, and the damage is concentrated in the energy dissipation ring, the damage is controllable, and the rapid repair after the earthquake can be realized by replacing energy dissipation devices and other measures. At the same time, the seismic performance of pier models with three different control parameters (initial prestress, material strength of energy dissipation ring and section width of energy dissipation ring) under reciprocating loading is analyzed. The results show that the initial prestress does not affect the cumulative energy consumption of the pier; the increase in the material strength of the energy dissipation ring improves the overall stiffness and improves the energy dissipation capacity of the PSCFST pier; the reduction in section width will affect the overall equivalent stiffness and unloading stiffness of the segmental pier, and the energy dissipation capacity will be significantly reduced.

**Keywords:** bridge engineering; precast segmental concrete-filled steel tubular bridge piers; energy dissipation ring; post-earthquake repair; reciprocating loading; seismic performance

## 1. Introduction

A prestressed segmental assembled pier can effectively reduce the residual displacement after an earthquake due to its self-centering ability. It can also greatly improve the construction efficiency because of the assembly property. However, the pier also has its disadvantages: the joint at the pier bottom section is seriously damaged under the action of an earthquake, and the energy dissipation capacity is low in earthquakes, so it is difficult to make popular in high-intensity areas.

In recent years, a lot of relevant research has been conducted on these problems. With the improvement in segmental connection and assembly technology, some precast pier

structures with high vertical bearing capacity and suitable for high-intensity areas have been proposed and studied. The segmental precast concrete-filled steel tubular pier is one of them. Due to the strong axial bearing capacity of concrete-filled steel tubular, scholars both domestic and foreign apply it to segmental assembled piers and a lot of research has been conducted [1], which has important guiding significance and great development potential for the popularization of segmental assembled piers in high-intensity areas. Concrete-filled steel tubular pier columns have reliable vertical and horizontal bearing capacity [2]. During the prefabrication, the steel pipe can become the side formwork of concrete, which has good economic benefits and constructability. The stress concentration of bonded prestress can be reduced by adopting unbonded prestress. At the same time, it can also avoid the local crushing of concrete caused by the penetration of prestress between segments [3]. At present, there are relatively many, mature studies on the performance of concrete-filled steel tubular structures, while the application of concrete-filled steel tubulars in segmental assembled piers and their performance need more thorough and detailed research [4].

Hewes [5] et al. conducted the test of wrapping the pier body with steel pipe in the plastic hinge area at the bottom of the pier, in order to give full play to the advantages of a concrete-filled steel tube and prevent the damage of the pier in the plastic hinge area. However, this practice leads to the plastic hinge moving up to the joint of the upper section, causing serious damage. Chou and Chen [6] replaced all reinforced concrete sections with concrete-filled steel tubular sections and added damping energy dissipation devices at the joint joints. The test shows that this kind of pier has good flexural performance and ductility, does not bring large residual displacement, and the energy dissipation is also improved. Zhou Shufen [7,8] took advantage of a series of advantages, such as high bearing capacity and good seismic performance of concrete-filled steel tubular piers, combined with the stress characteristics of medium and low piers of long-span short pier continuous rigid frame bridges, and takes Xi'an Weihe River Bridge as the engineering background, as the substructure adopts concrete-filled steel tubular piers. It is found that concrete-filled steel tubular piers can greatly reduce the seismic response of the structure, and have better ductility and energy dissipation capacity than concrete piers. Therefore, this kind of pier can be used as an ideal form of bridge pier column in a high-risk earthquake area. Jia Junfeng [9,10] introduced the design method of the main components of the prestressed segmental assembled concrete-filled steel tubular pier, and carried out the pseudo-static test of this kind of pier. It is found that this kind of pier has poor energy dissipation capacity without an additional energy dissipation device, and the addition of a connecting steel pipe at the segmental assembly joint can improve the energy dissipation effect, but increases the residual displacement of the pier top.

In recent years, the development of seismic engineering research in China has shown a trend from earthquake resistance, vibration reduction and isolation to recoverable function [11]. Therefore, for segmental precast assembled piers to gain attention and be improved, their serviceability and repairability after an earthquake should also be further improved and explored. ElGawady [12] et al. conducted a quasi-static test on a double-column self-centering precast pier with an angle steel damper. The research shows that the energy dissipation capacity of the precast assembled pier is increased by 75% compared with that of a precast assembled pier without an angle steel damper; when the offset rate is 4%, the residual displacement is about 10%. Later, due to the fracture in the angle steel damper, the energy dissipation capacity of the pier is reduced, and the concrete pier body is basically free of damage. Sun Zhiguo [13] et al. proposed a swing self-centering prefabricated assembled double-column pier with angle steel and energy dissipation reinforcement, and analyzed its seismic response under a near-fault earthquake based on the OpenSees numerical platform. The results show that the developed swing self-centering prefabricated assembled double-column pier has good seismic capacity and small residual displacement after an earthquake. Wang [14] et al. set replaceable energy dissipation devices in the plastic hinge area of an ultra-high-performance concrete (UHPC) hollow segment pier, and formed a replaceable energy dissipation reinforcement system by assembling UHPC slab.

The test results show that the pier damage is mainly concentrated in the replaceable system, and the pier still has similar seismic performance after replacing the energy dissipation device and UHPC plate. Although the above scholars conducted a series of studies on the self-centering pier of a replaceable energy dissipation system, the feasibility study of replaceable energy dissipation systems needs to be improved.

In view of the above problems, this paper proposes to use the external energy dissipation ring to improve the overall seismic performance of the prefabricated pier and realize rapid repair after the earthquake, establishes a finite element analysis model with ABAQUS, and discusses the feasibility of the external energy dissipation ring PSCFST pier by comparing with the seismic performance of the traditional PSCFST pier; the effects of initial prestress, material strength and section width of the energy dissipation ring on the seismic performance of a PSCFST pier with an external energy dissipation ring are analyzed.

## 2. Verification with Finite Element Model

### 2.1. Test Introduction

In order to verify the effectiveness of the established finite element analysis model, the axial compression test for a concrete-filled steel tubular column is carried out, and the finite element model of the test concrete-filled steel tubular column is established. The numerical analysis results are compared with the test results to verify the feasibility of the finite element model.

The cross-sectional dimension of the test piece is 230 × 230 mm, height 700 mm. Q235 steel is used for the steel pipe and the concrete is C40. The connecting part between the short column and the foundation is directly anchored, and the loading mode adopts displacement control. The dimensions of the test piece are shown in Figure 1.

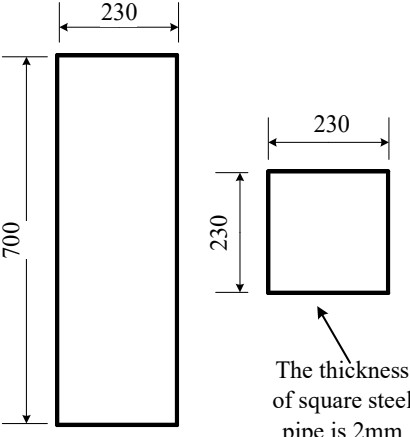

**Figure 1.** Specimen size of concrete-filled steel tubular column (mm).

Tensile tests on steel coupons were conducted based on the Chinese Standard GB/T 228-2010 [15]. Each set of tests consisted of three tensile specimens and the results were averaged. The measured yield strength and elastic modulus are listed in Table 1, where $f_y, f_u, E_s$ represents the yield strength, ultimate tensile stress and elastic modulus of steel. Three concrete blocks (150 mm cube) were tested under axial loading in each group to obtain the compressive strength, and the results were averaged. According to the conversion relationship between the prism compressive strength ($f_c$) and the cubic compressive strength ($f_{cu}$) suggested in Chinese Standard GB 50010-2010 [16], the compressive strength of the concrete prism was calculated and is listed in Table 2, where $E_c$ represents the elastic modulus of concrete.

**Table 1.** Material properties of steel.

| Steel | $f_y$/MPa | $f_u$/MPa | $E_s$/$10^5$ MPa |
|---|---|---|---|
| Q195 | 201 | 284 | 2.01 |
| Q235 | 247 | 321 | 2.02 |
| Q345 | 363 | 542 | 1.96 |

**Table 2.** Material properties of concrete.

| Concrete | $f_{cu}$/MPa | $f_c$/MPa | $E_c$/$10^4$ MPa |
|---|---|---|---|
| C40 | 26.8 | 18.4 | 2.26 |

Figure 2a shows the steel pipe concrete column specimen before loading. With the larger load on the concrete-filled steel tubular column, the test specimen experiences the elastic stage, peak bearing capacity stage, and failure stage. When the bearing capacity of the concrete-filled steel tubular short column reaches the peak, the part circled by the red line in Figure 2b shows no obvious drum. As the load increases, the damage in the concrete-filled steel tubular column becomes more and more obvious, and the drum of the steel tube becomes more and more obvious.

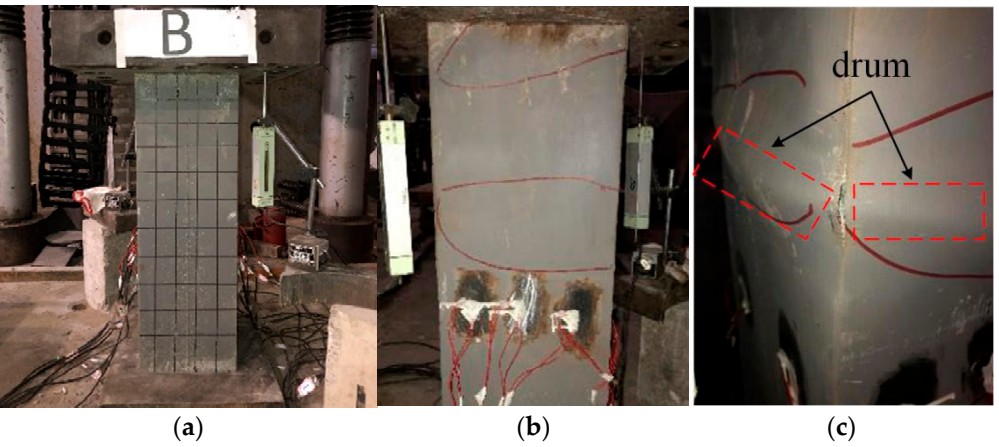

| (**a**) | (**b**) | (**c**) |

**Figure 2.** Concrete-filled steel tubular specimen. (**a**) Before test loading. (**b**) Peak bearing capacity. (**c**) After test loading.

With the increase in load, the bearing capacity of the steel pipe is lost. It can be seen that the bulge of the steel pipe stopped after the vertical loading (as shown in Figure 2c).

### 2.2. Constitutive Relation

In this study, the steel pipe finite element model is simulated with the various isotropic elastic-plastic models provided by ABAQUS, and the stress–strain is simulated with the ideal elastic-plastic model, as shown in Figure 3. The constitutive relationship expression is as follows:

$$\sigma = \left\{ \begin{array}{l} E_s\varepsilon, \varepsilon \leq \varepsilon_y \\ f_y , \varepsilon > \varepsilon_y \end{array} \right\} \tag{1}$$

where:

$E_s$—elastic modulus of steel;
$f_y$, $\varepsilon_y$—yield strength and corresponding yield strain of steel;
$\sigma$, $\varepsilon$—steel stress and corresponding strain.

The elastic modulus of square steel pipe is $2.02 \times 10^5$ MPa, Poisson's ratio is 0.25, and the yield strength is 247 MPa.

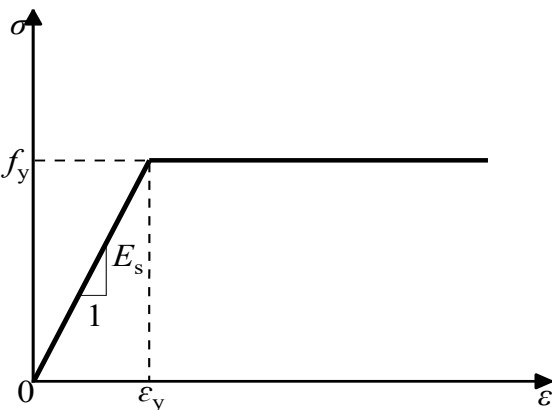

**Figure 3.** Stress–strain curves of steel.

The concrete damage plasticity model (CDP) is based on the principles of damage mechanics and considers the effects of damage on concrete materials under monotonic loading, reciprocal loading and dynamic analysis. The CDP is based on the principles of damage mechanics and considers the effects of damage generation, accumulation and evolution on the material under monotonic loading, reciprocal loading and dynamic analysis of loading conditions. It is suitable for simulating the stiffness degradation of the material due to damage and the mechanical behavior of stiffness recovery during reverse loading. It is suitable for modelling the mechanical behavior of the material in terms of stiffness degradation due to damage and stiffness recovery during reverse loading. Based on this, the CDP model is used to establish the intrinsic structure relationship for concrete sections. The concrete plastic damage parameters are shown in Table 3.

**Table 3.** Plastic damage parameters for concrete.

| Concrete | Expansion Angle | Flow Potential Bias | Stress Ratio | Invariant Stress Ratio | Viscosity Coefficient |
|---|---|---|---|---|---|
| C40 | 35 | 0.1 | 1.16 | 0.667 | 0.0005 |

Core concrete in steel pipes under axial compression and other external loads is a typical confined concrete under three-way compression. The constitutive relation of section concrete in this paper adopts the restraint concrete constitutive model proposed by Han Linhai [17], and its stress–strain curve is shown in Figure 4.

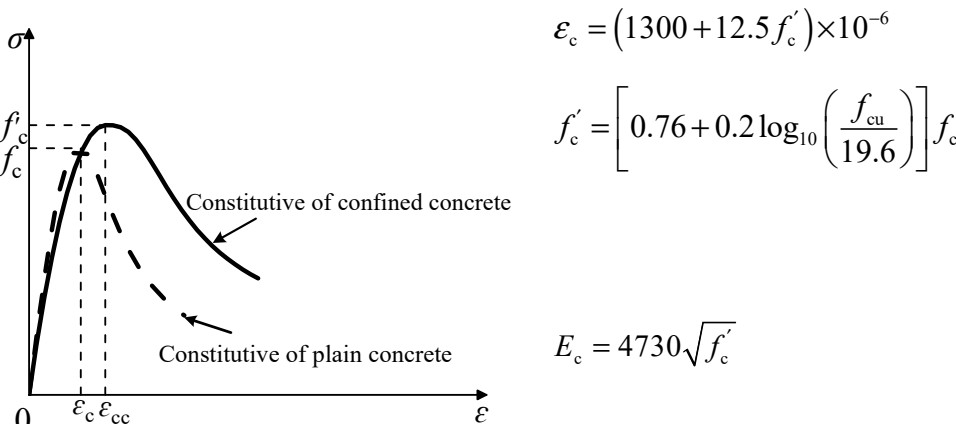

$$\varepsilon_c = \left(1300 + 12.5 f_c'\right) \times 10^{-6}$$

$$f_c' = \left[0.76 + 0.2 \log_{10}\left(\frac{f_{cu}}{19.6}\right)\right] f_{cu}$$

$$E_c = 4730\sqrt{f_c'}$$

**Figure 4.** Stress–strain curves of concrete.

### 2.3. Model Validation

Figure 5 shows the load displacement curve comparison between the test results of concrete-filled steel tubular short columns and the finite element results. It can be seen that the curves are in good agreement as a whole, and the elastic stiffness of the curve obtained by the finite element is greater than the test results, which is due to the fact that the influence of initial defects, residual stress and other factors of the member are not considered in the process of finite element analysis. The ratio of the axial compression bearing capacity obtained from the finite element and the test is 1.011, and the difference of the bearing capacity is less than 2%, which is in good agreement on the whole.

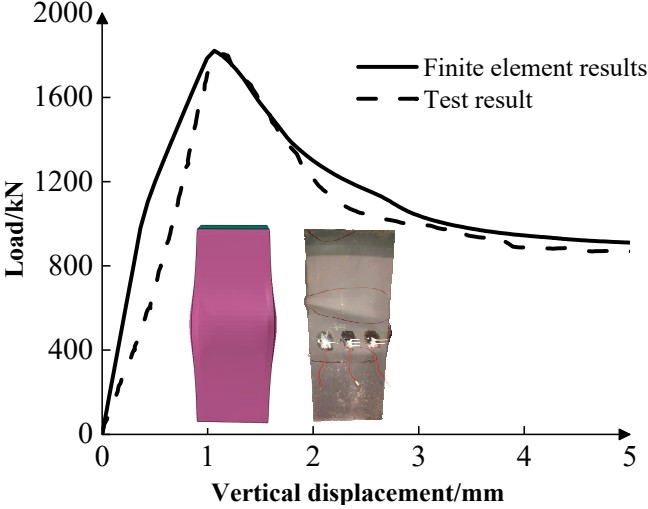

**Figure 5.** Load-displacement curve.

Figure 6 show the load transverse strain curve and load longitudinal strain curve of concrete-filled steel tubular short columns under test and finite element analysis, respectively. According to Figure 6, the ratio of the finite element to the lateral limit strain on the left side of the test specimen is 1.012, the ratio of the lateral limit strain on the right side is 1.008, and the average value is 1.01. The match of the curve is good and reliable. Similarly, according to Figure 6, the ratio of the finite element to the longitudinal limit strain on the left side of the test specimen is 1.013, the ratio of the longitudinal limit strain on the right side is 1.011, and the average value is 1.012.

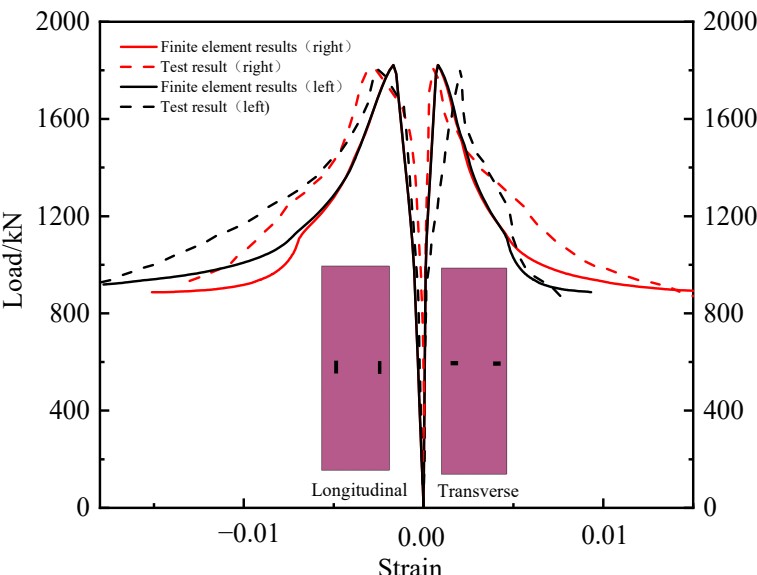

**Figure 6.** Load transverse strain curve and load longitudinal strain curve.

In general, the finite element results are in good agreement with the test results and the errors are within a reasonable range, showing good reliability. It seems that the constitutive relationship and parameter setting adopted in this paper can accurately simulate the concrete-filled steel tubular structure, which can be the base for the following simulation.

## 3. Establish Finite Element Numerical Model

### 3.1. Specimen Design and Loading Scheme

Based on the traditional PSCFST pier, a PSCFST pier with an external replaceable energy dissipation ring is designed. The traditional PSCFST pier and the PSCFST pier with the external energy dissipation ring are named specimen A and specimen B, respectively. The test piece is composed of a bearing platform foundation, pier body and pier cap. The pier body of the precast assembled pier is divided into four segments: S1, S2, S3 and S4. The height of each segment is 334 mm. The rectangular section is adopted, the section size is 230 mm, the effective pier height is 1486 mm, and the axial compression ratio is 0.4 [18]. The pier cap and the bearing platform are connected by four 15.2 mm steel strands, the applied prestress is 103 kN (through calculation, we find that the ultimate bearing capacity of the PSCFST pier specimen is 1028 kN, so the working condition with initial prestress of 103 kN is adopted; that is, the axial compression ratio of the initial prestress is 0.1). The unbonded part is arranged in the center of the pier. A detachable energy dissipation ring member is set at the section interface, and the energy dissipation ring member and steel pipe section are fixed with a high-strength bolt cap. Q235 steel is used for the steel pipe section and energy dissipation ring, and the internal core concrete is C40 concrete. The structure of the test piece and the size of the energy dissipation ring are shown in Figure 7.

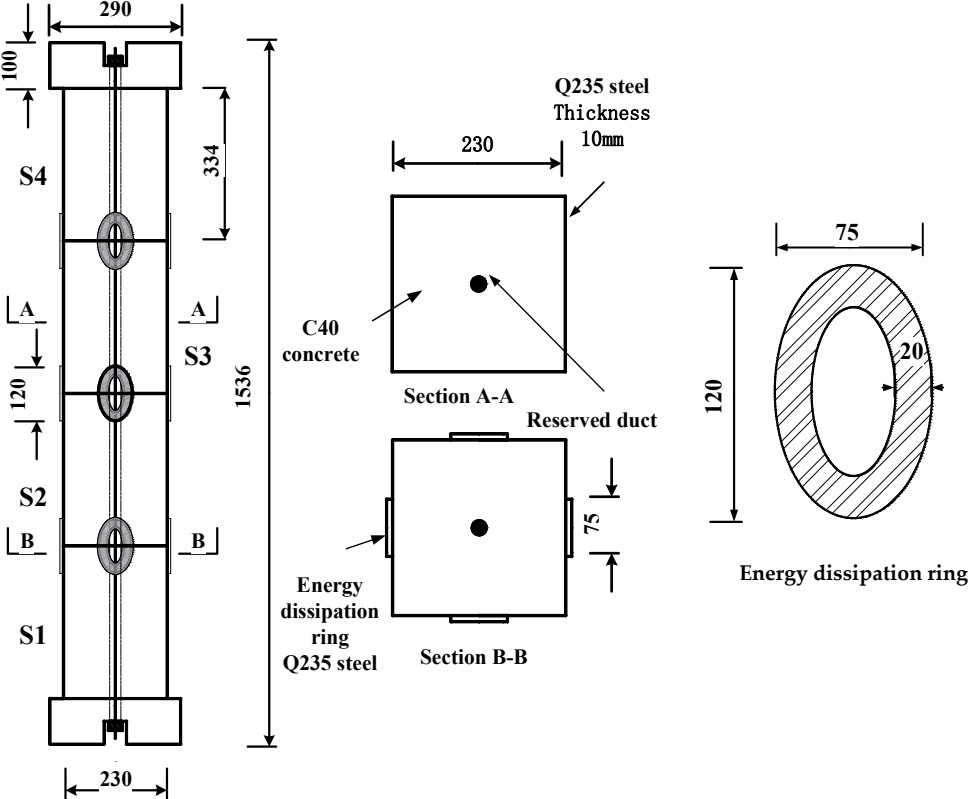

**Figure 7.** Structure of precast assembled pier test piece (unit: mm).

For the traditional PSCFST piers, the size structure and prestressing tendon arrangement are the same as the PSCFST piers with external energy dissipation rings. The only difference is that the traditional PSCFST piers do not have external energy dissipation rings.

The specimen is loaded with low-cycle reciprocating loading, which is located in the center of the side of the pier cap, and the bottom of the pier column is fully restrained to form a cantilever structure. The loading mode adopts displacement control, the displacement to be loaded is 5 mm, 10 mm, 15 mm, 20 mm, 25 mm, 30 mm, 35 mm, 40 mm and the loading cycle is once at each stage. The loading curve is shown in Figure 8.

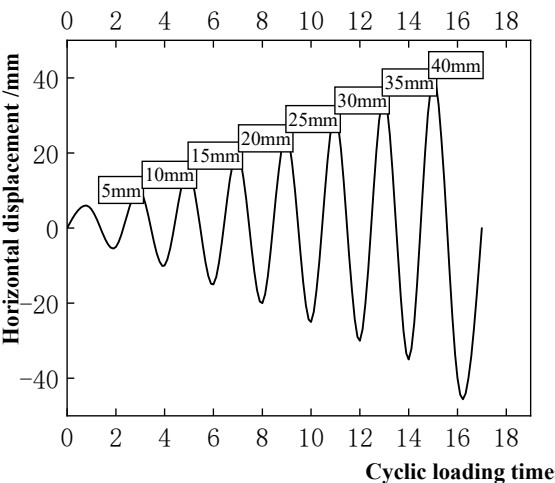

**Figure 8.** Displacement loading scheme.

### 3.2. Establishment of Finite Element Model

Figure 9 shows the finite element model of precast assembled pier, which is made up of steel pipe, concrete and energy dissipation rings. An eight-node linear hexahedron C3D8R element is used to simulate the concrete and steel pipe. The prestressed reinforcement is simulated by a truss element (T3D2). The contact segments adopt face-to-face contact. When the gap is zero, the pressure is transmitted along the normal direction of the contact surface. When the gap is greater than zero, the tension and pressure are no longer transmitted. Assuming the tangential direction between segments is contact friction, the friction coefficient $\mu$ is 0.6 [19]. The steel pipe and concrete are bound together by *Tie* contact.

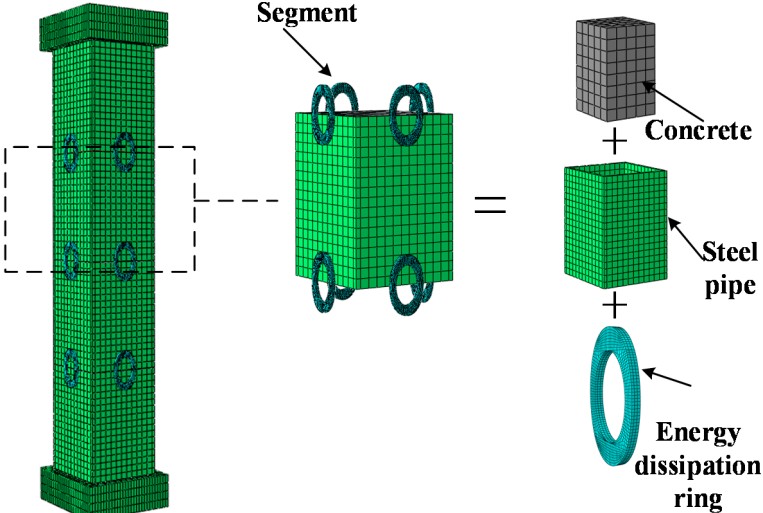

**Figure 9.** Finite element model of precast assembled pier.

In the actual project, the energy dissipation ring will be connected and fixed with the PSCFST pier through high-strength bolts and embedded screws in the section. In the finite element method, the energy dissipation ring and the steel pipe wall are in contact with *Tie* to bring the high-strength bolts.

For the convenience of analysis, the piers involved in this paper are designed as a cantilever structure, the bottom end of the pier is fixed, all degrees of freedom are restrained, and the top end of the pier is free, without any restraint. In order to obtain the reaction force at the bottom of the pier, a reference point is established in the center of the base, the coupling method is used to couple the base center and the reference point, and the translational and rotational degrees of freedom in three directions of the reference point are constrained, so as to achieve the consolidation effect of the pier bottom.

The prestressed reinforcement is designed as unbonded prestressed reinforcement. Therefore, the prestressed reinforcement is divided into three parts. The part of the prestressed reinforcement extending into the loading end is embedded and connected with the foundation, and the other parts are not treated to simulate the unbonded state of the prestressed reinforcement between the concrete. There are three steps in the analysis. The first step is to load prestress. Secondly, the gravity load and axial dead load pressure need to be defined. Finally, the lateral low-cycle reciprocating loading must be defined. The axial pressure is applied by the gravity of the counterweight block on the pier top, and the prestress is applied by the cooling method. The cooling amplitude should be set to 200 °C and the initial prestress should be set to 103 kN.

## 4. Finite Element Analysis

### 4.1. Hysteretic Curve and Skeleton Curve

Figure 10a shows the hysteretic curve of the traditional PSCFST pier and PSCFST pier with an external energy dissipation ring.

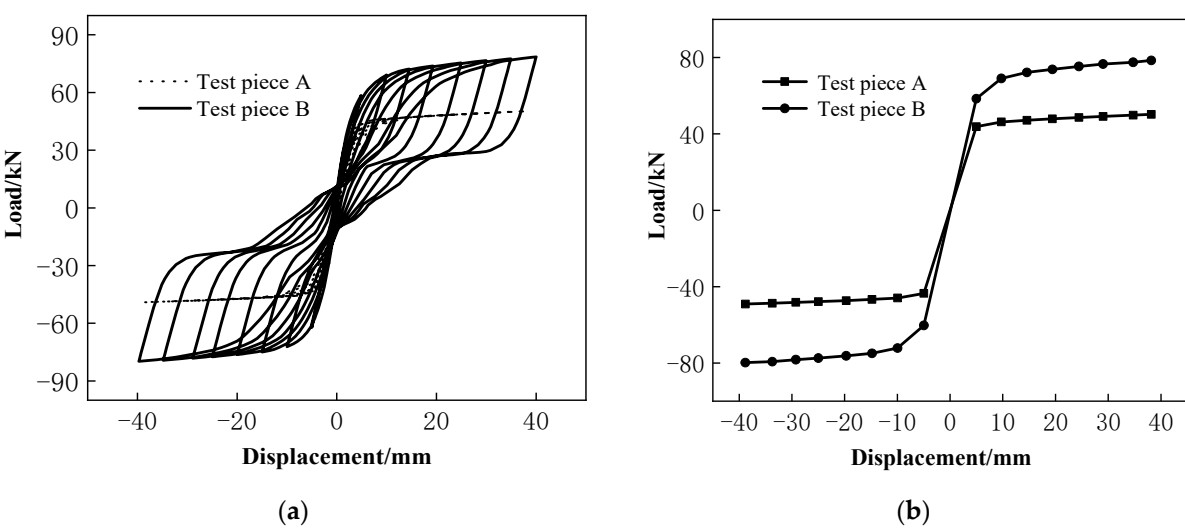

(**a**)          (**b**)

**Figure 10.** Comparison diagram of hysteretic characteristics. (**a**) Hysteretic curves of different piers. (**b**) Skeleton curves of different types of piers.

Comparing the hysteretic curves piers, it can be seen that:

(a) The hysteretic curve for the traditional precast segmental assembled concrete-filled steel tubular pier is similar to the flag shape, the hysteretic loop area is small, and the energy dissipation capacity of the pier is poor. The reloading stiffness and unloading stiffness are relatively stable, the stiffness degradation is not obvious, and the residual displacement is small, indicating that the self-centering capacity of the pier is good, and the maximum horizontal bearing capacity is 50 kN.

(b) The hysteretic curve of the segmental assembled concrete-filled steel tubular pier with an external energy dissipation ring is relatively full, the area of the hysteretic ring is significantly increased, and the energy dissipation capacity is significantly improved. Due to the influence of sliding between segments, the energy dissipation ring is sheared and deformed, and the hysteretic curve has an obvious pinch effect. Although the residual displacement increases slightly, it still maintains a small level. Since the energy dissipation

ring is set at the segment joint, the maximum horizontal bearing capacity is 79.6 kN, which is increased by about 60%.

Figure 10b shows the skeleton curve for the traditional PSCFST pier and PSCFST pier with an external energy dissipation ring. Table 4 shows the eigenvalues of the skeleton curve.

**Table 4.** Performance turning point of skeleton curve.

| Test Piece No | Yield Strength | Yield Displacement | Peak Bearing Capacity | Peak Displacement | Ductility Coefficient |
|---|---|---|---|---|---|
| A | 48.67 kN | 24.98 mm | 50.28 kN | 38.17 mm | 1.56 |
| B | 75.48 kN | 25.37 mm | 78.51 kN | 40 mm | 1.58 |

Comparing the skeleton curve and its characteristic value, it can be seen that the yield strength pier columns are 48.67 kN and 75.48 kN, respectively. The external energy dissipation ring can improve the initial stiffness and lateral bearing capacity of the PSCFST pier.

### 4.2. Cumulative Energy Dissipation

Figure 11 shows the cumulative energy dissipation curve for the traditional PSCFST pier and PSCFST pier with an external energy dissipation ring.

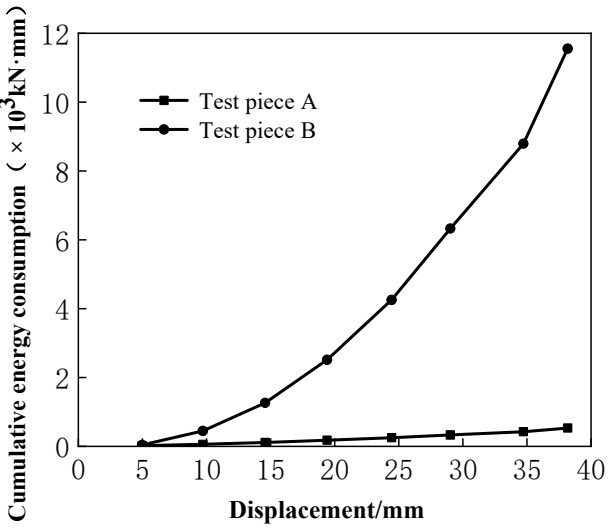

**Figure 11.** Cumulative energy dissipation curve of different types of piers.

Comparing the cumulative energy dissipation curves, it can be seen that the energy dissipation capacity of the PSCFST pier with an external energy dissipation ring is significantly improved compared with the traditional PSCFST pier; when the two specimens reach the maximum displacement, the cumulative energy dissipation of the traditional PSCFST pier is 528.9 kN·mm, and the energy dissipation capacity of the PSCFST pier with the external energy dissipation ring is 11,555.4 kN·mm, about 20-times larger than the former. The energy dissipation ring can continue to play a role in the loading process.

### 4.3. Plastic Energy Dissipation and Friction Energy Dissipation

Figure 12 shows the plastic energy dissipation curve and friction energy dissipation curve of the traditional PSCFST pier and PSCFST pier with an external energy dissipation ring.

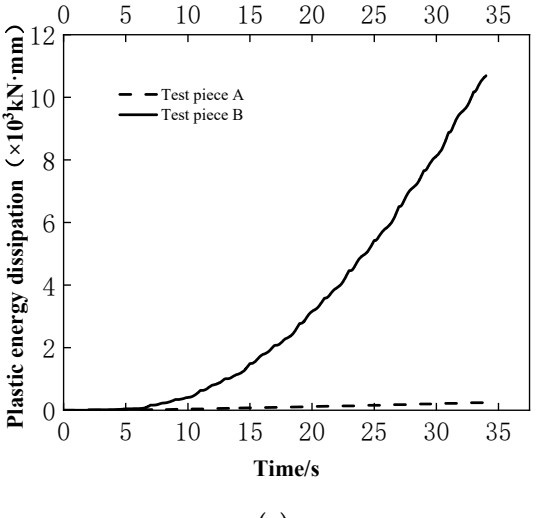
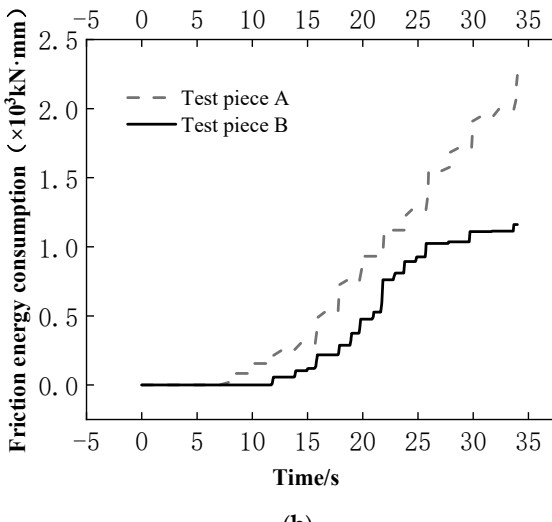

**Figure 12.** Comparison of friction energy consumption and plastic energy consumption curves. (**a**) Plastic energy dissipation curve. (**b**) Friction energy dissipation curve.

Plastic energy dissipation refers to the hysteretic energy dissipation caused by the plastic strain of the components in the pier in the hysteretic reciprocating motion of the pier; friction energy dissipation is the energy dissipated due to the friction of each part in the low-cycle reciprocating motion of the pier. In the ABAQUS finite element model mentioned in this paper, only the friction contact surface is set at the joints of concrete-filled steel tubular segments. Therefore, in this paper, it can be considered that the friction energy dissipation refers to the energy generated by the friction between segments.

As can be seen from Figure 12a, compared with the traditional PSCFST pier, the PSCFST pier with an external energy dissipation ring mainly relies on plastic energy dissipation for energy dissipation; that is, the whole pier mainly depends on the yield deformation of the energy dissipation ring itself to achieve energy dissipation.

It can be seen from Figure 12b that there is a great difference between the mechanisms of friction energy dissipation; the overall friction energy dissipation of the traditional PSCFST pier is higher than that of the PSCFST pier with an external energy dissipation ring. The reason is that the energy dissipation ring can effectively protect the joint surface of the concrete-filled steel tubular section, prevent the dislocation between the concrete-filled steel tubular sections, and keep the damage in the concrete-filled steel tubular column at the joint low, thus achieving the aim of protecting the steel pipe concrete at the section joints.

*4.4. Plastic Strain*

Figure 13 shows the equivalent plastic strain nephogram of the PSCFST pier with an external energy dissipation ring and the traditional PSCFST pier.

Comparing the equivalent plastic strain nephogram, it can be seen that the steel pipe part of the traditional PSCFST pier does not have plastic strain and does not participate in energy dissipation; the plastic strain on the PSCFST pier with an external energy dissipation ring is concentrated in the energy dissipation ring, which shows that the external energy dissipation ring can improve the energy dissipation capacity of the PSCFST pier and concentrate the damage on the energy dissipation ring. The deformation and energy dissipation of the pier are mainly concentrated at the joint, and there will be no excessive joint opening and serious damage in the middle and upper part of the pier body. Both the energy dissipation ring along the loading direction and the energy dissipation ring on the other side have plastic strain, which can play the role of energy dissipation, while the deformation of the energy dissipation ring along the loading direction is more obvious, indicating that its energy dissipation contribution rate is higher, and the lateral stiffness of the pier is effectively improved. In addition, the energy dissipation ring at the bottom joint

is only damaged in the middle, while the joints at both ends are still in the elastic stage, which ensures that the energy dissipation ring is easy to replace after the earthquake.

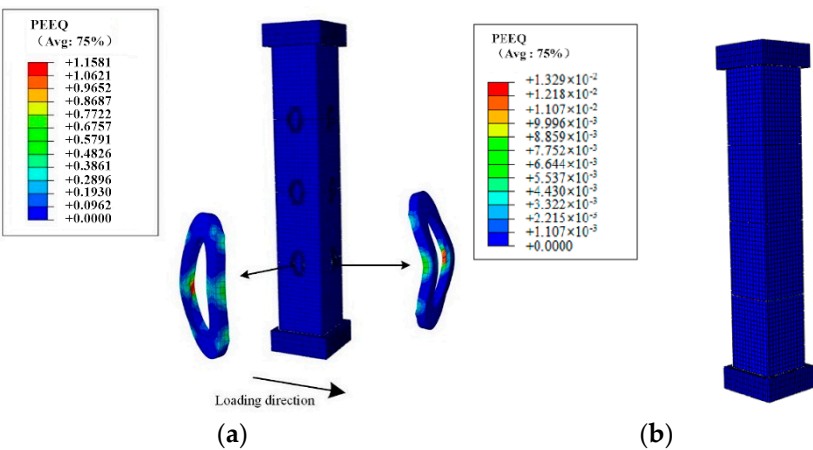

**Figure 13.** PEEQ (equivalent plastic strain nephogram). (**a**) PSCFST pier with external energy dissipation ring. (**b**) Traditional PSCFSt pier.

## 5. Analysis on Influence Factors of Seismic Performance

The factors affecting the seismic performance of segmental assembled concrete-filled steel tubular piers, such as prestress, reinforcement ratio, axial compression ratio and section steel content, have been explored in previous research. In order to verify the improvement in the energy dissipation capacity of the energy dissipation ring on the segmental assembled concrete-filled steel tubular pier, this section mainly explores the effect of the initial prestress, the material strength of the energy dissipation ring and the section width of the energy dissipation ring on the PSCFST pier.

### 5.1. Initial Prestress

Prestressed tendons in PSCFST piers can protect the self-centering capacity of piers, which are usually considered to be energy dissipation components. Therefore, the prestress size is also taken into account to explore the seismic performance of PSCFST piers when the initial prestress size is 52 kN, 103 kN and 155 kN, respectively. Figure 14 shows the hysteretic curve and skeleton curve of the PSCFST pier under different prestress.

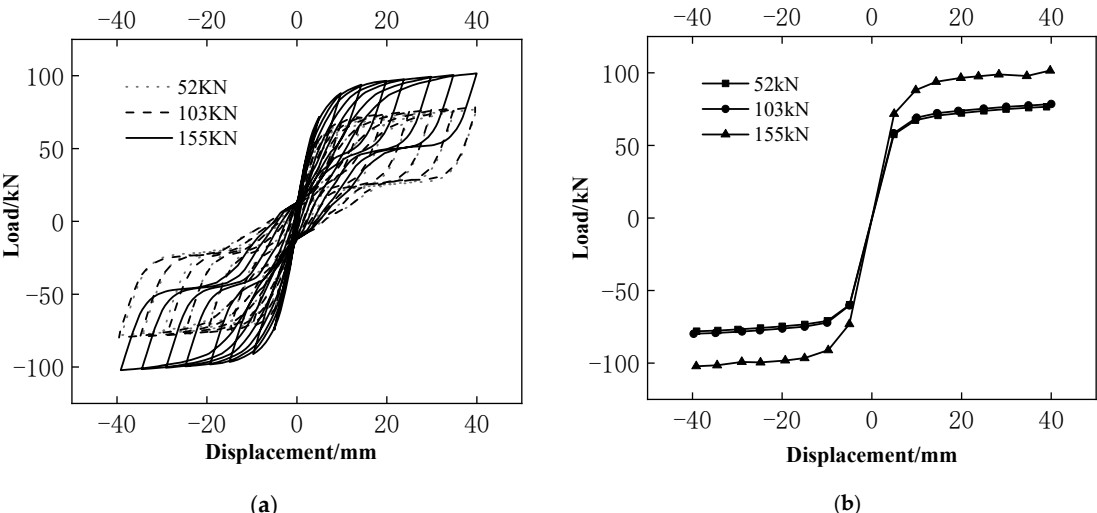

**Figure 14.** Comparison diagram of hysteretic characteristics. (**a**) Hysteretic curve under different prestress. (**b**) Skeleton curve under different prestress sizes.

Comparing the hysteretic curve and skeleton curve of the PSCFST pier under different prestress, it can be seen that when the initial prestress is 52 kN and 103 kN, the curve trend tends to be the same. When the maximum displacement is 38.9 mm, the horizontal force is 76.6 kN and 78.5 kN, respectively. When the initial prestress increases to 155 kN, it can be seen that the curve changes. When the maximum displacement is 39.7 mm, the horizontal force is 101.6 kN. It may be that the initial prestress is increased to 155 kN, which makes the connection between segments closer. Due to the extrusion of prestress, the steel pipe and concrete can more effectively participate in the reciprocating motion, which improves the lateral bearing capacity of the PSCFST pier.

Compared with the cumulative energy dissipation curve of the pier (as shows in Figure 15), it can be seen that although the lateral bearing capacity of the PSCFST pier has been improved, the energy dissipation capacity in the three groups of models is the same, and the area of the energy dissipation ring is basically the same. It can be inferred that the size of initial prestress will not affect the energy dissipation capacity of the PSCFST pier.

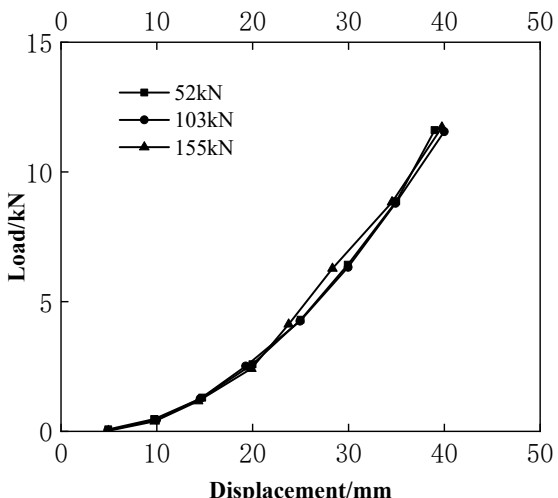

**Figure 15.** Cumulative energy dissipation curve under different prestress sizes.

Compared with the friction energy dissipation curves of PSCFST piers under different prestress sizes (as shows in Figure 16), it can be obtained that with the increase in prestress, the friction energy dissipation of the piers has increased significantly, indicating that the increase in prestress will improve the integrity of piers and increase the friction between segmental joints, so that piers can dissipate more energy under the same displacement conditions.

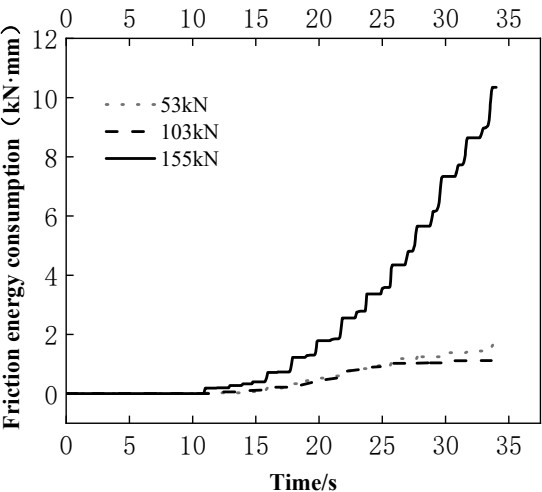

**Figure 16.** Friction energy dissipation curve under different prestress.

### 5.2. Material Strength of Energy Dissipation Ring

When the axial compression ratio of prestress is 0.05, the initial prestress is 52 kN; when the axial compression ratio of prestress is 0.15, the initial prestress is 155 kN.

Comparing the hysteretic curve and skeleton curve of the PSCFST pier with different material strengths of the energy dissipation ring (as shows in Figure 17), it can be found that with the increase in material strength of the energy dissipation ring from Q195 to Q345, the hysteretic curve and skeleton curve of the pier gradually increase, with a more obvious flag shape. This is due to the increase in the strength of the connecting members between segments, which makes the integrity of the pier better, the ultimate bearing capacity of the pier gradually increases, and the overall stiffness of the pier has been significantly improved.

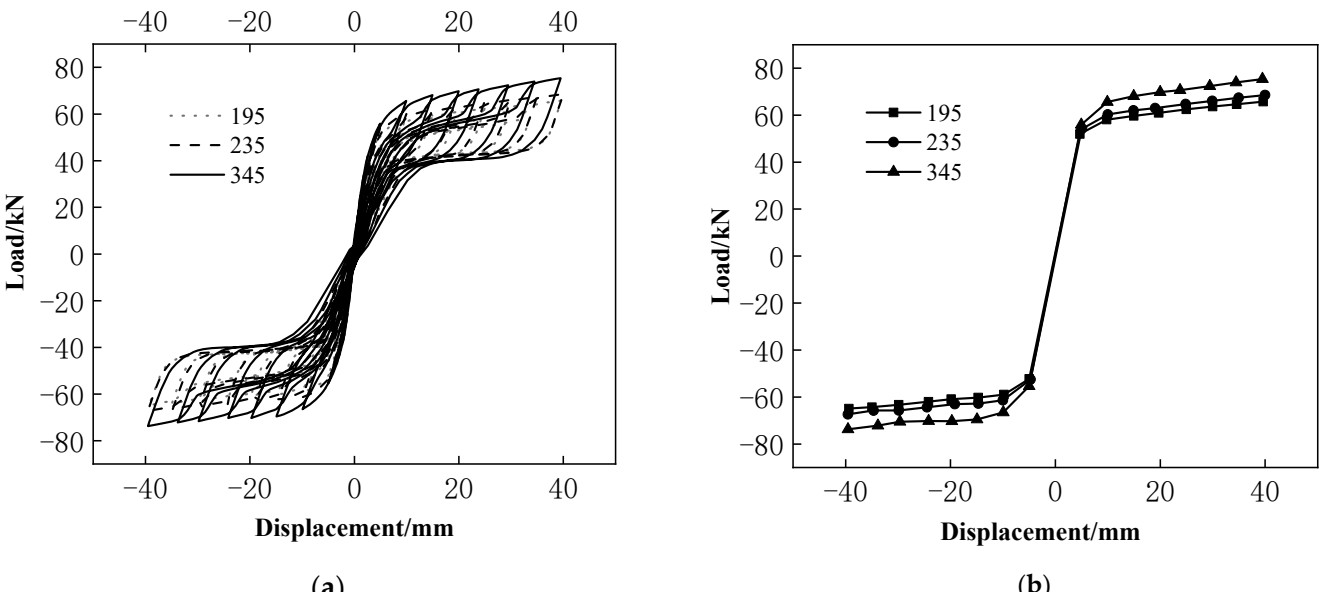

**Figure 17.** Hysteretic curve and skeleton curve under different energy dissipation ring strength. (**a**) Hysteretic curve. (**b**) Skeleton curve.

In the elastic stage of the first loading, it can be seen that the hysteretic curves and skeleton curves of PSCFST piers with different material strengths of energy dissipation rings basically coincide, indicating that the overall stiffness has nothing to do with the material strength when it does not enter the damage stage at the beginning. The slip section of unloading is basically the same, indicating that the influence of material strength on unloading stiffness is also relatively small.

As shows in Figure 18, with the increase in the material strength of the energy dissipation ring, the energy dissipation capacity of the PSCFST pier has been significantly improved, which can be seen from the final cumulative energy dissipation: Q195 corresponds to 3726.7 kN·mm and Q345 corresponds to 5036.45 kN·mm, up 26 percent.

Comparing the friction energy dissipation curves of PSCFST piers with different energy dissipation ring material strengths (as shows in Figure 19), it can be seen that the friction energy dissipation of piers is inversely proportional to the material strength within 20 s before loading, but with the increase in energy dissipation ring damage, the friction in the concrete-filled steel tubular sections between segments gradually increases. In the later stage of loading, it can be seen that the friction energy dissipation in the model with the Q345 energy dissipation ring material increases faster. With the increase in the material strength of the energy dissipation ring, the friction loss also increases.

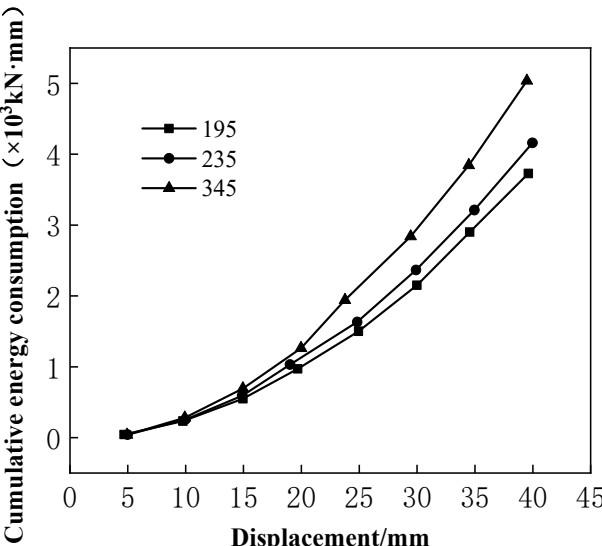

**Figure 18.** Cumulative energy dissipation curve under different energy dissipation ring strengths.

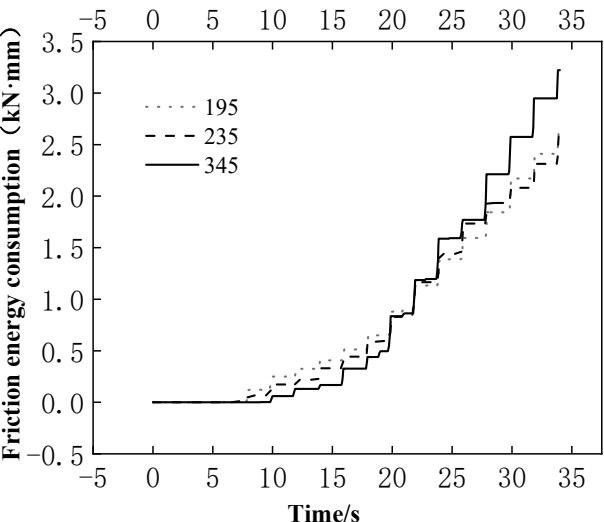

**Figure 19.** Friction energy dissipation curve under different energy dissipation ring strengths.

### 5.3. Section Width of Energy Dissipation Ring

Comparing the hysteretic curves, skeleton curves and cumulative energy dissipation curves of PSCFST piers with different energy dissipation ring section widths (as shows in Figures 20 and 21), it can be seen that when the energy dissipation ring section width is 20 mm, the lateral load corresponding to the horizontal displacement of 40 mm is 78.51 kN; when the energy dissipation ring section width is 10 mm, the lateral load corresponding to the horizontal displacement of 40 mm is 63.51 kN, which is reduced by 19.1%, and the overall stiffness of the pier is reduced. When the section width is 10 mm, the residual displacement of the PSCFST pier is small, and the overall pinch phenomenon is obvious, but the hysteresis loop is not full enough, and the energy dissipation capacity is significantly reduced on the premise of stable stiffness development.

Comprehensive analysis shows that although the increase in initial prestress does not affect the energy dissipation capacity of PSCFST piers, when the initial prestress increases to a certain value, the lateral bearing capacity and initial stiffness of PSCFST piers will be significantly increased, so increasing the prestress degree will increase the ability of PSCFST piers to resist deformation. The increase in the energy dissipation ring strength makes the overall stiffness of the PSCFST pier increase, the ultimate bearing capacity of the pier also rises gradually, the hysteresis curve is fuller, the energy dissipation capacity

of the pier also increases gradually, but at the same time, with the increase in the energy dissipation ring strength, the residual displacement of the pier increases slightly. The width of the energy dissipation ring section affects the lateral bearing capacity of the PSCFST pier, with a decrease in the width of the energy dissipation ring section, the lateral bearing capacity of the pier also decreases gradually, and the energy dissipation capacity will also decrease and the phenomenon of low-energy dissipation capacity appears.

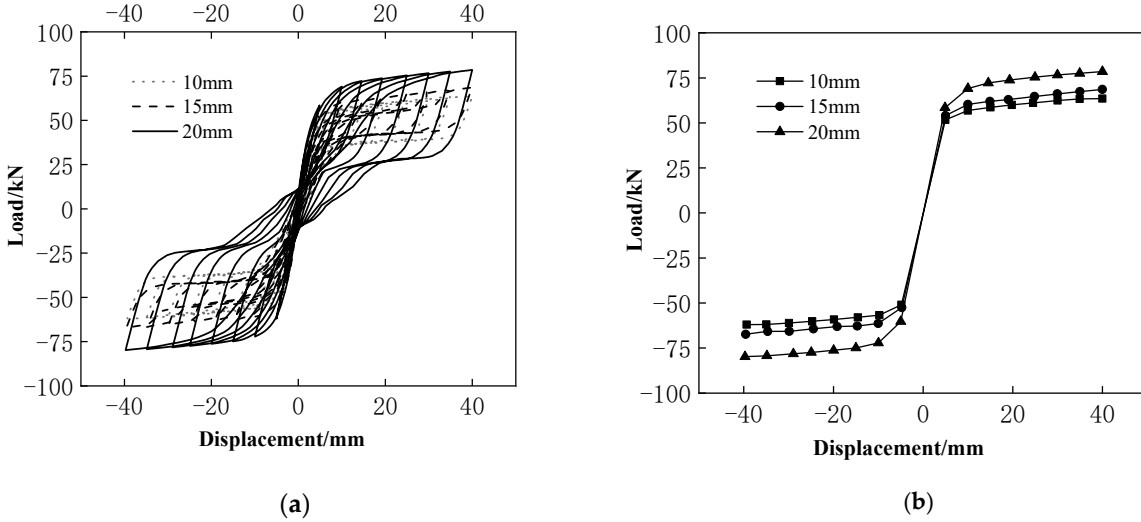

(**a**)    (**b**)

**Figure 20.** Hysteretic curve and skeleton curve under different section widths of energy dissipation ring. (**a**) Hysteretic curve. (**b**) Skeleton curve.

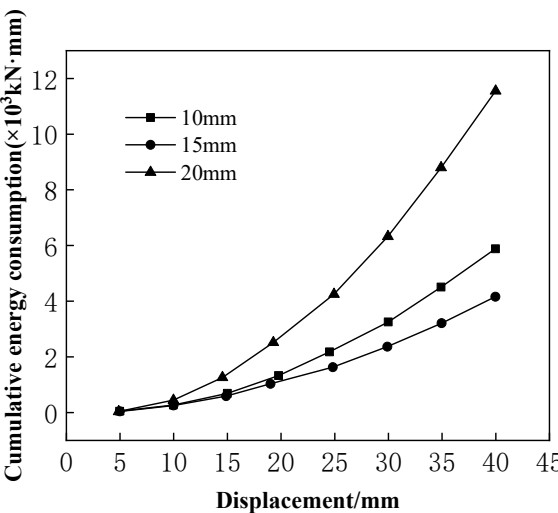

**Figure 21.** Cumulative energy dissipation curve under different section widths of energy dissipation ring.

## 6. Conclusions

(1) The hysteretic curve for the PSCFST pier with an external energy dissipation ring is fuller, and the energy dissipation capacity is greatly improved compared with the traditional PSCFST pier. The pier can accumulate more energy released by an earthquake. Although the residual displacement increases slightly, it remains at a small level. Therefore, the external energy dissipation device is necessary to improve the seismic capacity of the PSCFST pier.

(2) The PSCFST pier with an external energy dissipation ring mainly relies on plastic energy dissipation for energy dissipation; that is, the whole pier mainly depends on the yield deformation of the energy dissipation ring itself to achieve energy dissipation. The damage can be concentrated in the energy dissipation ring elements.

(3) The change in initial prestress will not affect the energy dissipation capacity of the PSCFST pier, but will increase its initial stiffness and resistance to deformation. With the improvement in the material strength of the energy dissipation ring, the energy dissipation capacity of the PSCFST pier is gradually improved, and the lateral bearing capacity is also significantly improved. The section width of the energy dissipation ring will affect the energy dissipation capacity of the PSCFST pier (the wider the section of the energy dissipation ring, the stronger the energy dissipation capacity of the PSCFST pier). Therefore, in the later stage, we can improve the overall capacity of the PSCFST pier by optimizing the energy dissipation ring.

**Author Contributions:** The innovation of the article and the idea of writing the first draft were proposed by C.W., Y.Z. (Yun Zou) provided many suggestions and helped revise the paper. Z.Q. carried out the experimental design and data analysis of the paper and substantially contributed to writing and revising the paper. C.Y., Y.Z. (Yanwei Zong) and Z.S. provided substantial help in preparing relevant data, finite element analysis, and the contents of the paper in the early stages. All authors have read and agreed to the published version of the manuscript.

**Funding:** This research received no external funding.

**Institutional Review Board Statement:** Not applicable.

**Informed Consent Statement:** Not applicable.

**Data Availability Statement:** All data, models, or codes that support the findings of this study are available from the corresponding author upon reasonable request.

**Conflicts of Interest:** The authors declare no conflict of interest.

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
