# Peer review of "Seismic Performance Analysis of Segmental Assembled Concrete-Filled Steel Tubular Pier with External Replaceable Energy Dissipation Ring"

_applsci, doi:10.3390/app12094729_

Round 1
Reviewer 1 Report
ID: applsci-1662992
Title: Seismic Performance Analysis of Segmental Assembled Concrete Filled Steel Tubular Pier with External Replaceable Energy Dissipation Ring
This paper proposed a new prefabricated assembled pier system having external energy dissipation ring. Authors have performed both numerical analysis and experimental test on the pier system. Owing to the ring, the proposed pier system has a good seismic performance. Additional numerical simulation using ABAQUS has been conducted such as initial prestresses, material strength of energy dissipation ring, and section width of energy dissipation ring.
There are some issues to overcome for publication:
- There are too many figures. It should be reduced.
- There is no conclusion. Only summary of this work presented in the conclusion section.
- There is too ordinary explanation for results of numerical simulations. Please, condense and describe only the important facts.
Major :
Line 14: what is full name of PSCFST? It seems that CFST means Concrete Filled Steel Tubular. What is PS? Is it prestressing segments? Please check it.
Line 124, 125, 132: There is no label for figures. Please insert label for each picture and merge them if possible. Also, No reference for Figure 2. Please check it.
Line 136 : ‘Constitutive relation’ Section, Please provide more information of the model and its parameters.
Line 144 : There is no ‘f’, Is it a typo ?
Line 145: 2.0GPa is not a common value for the steel elastic modulus. Please check it
Line 160: what kind of material model and its parameters for concrete material in ABAQUS? Please provide detailed information.
Line 171,173,176 : please merge three figures into one and labels them if possible. Also, use different line styles for the results of left and right side.
Line 177~184: what is the main fact having different results of numerical simulations for left and right side?
Line 200: What is the meaning of 103 kN? How did you select this value?
Line 225: for 0.6 friction coefficient. How did you select this value? Please provide the information how to get this friction coefficient.
Line 352: What is the meaning of 52,103,and 155 kN? How did you select this value?
Line 437 : Conclusion Section, there is no conclusion. Only summary of this work presented in the conclusion section. Please provide the conclusion of your work.
Minor:
Line 42 : some Precast Pier -> some precast pier, Is there any meaning on using upper case?
Line 52 : Unboned Prestressed. Same thing.
Line 76: Please remove [9]. It was cited doubly.
Line 84: ElGawady [12] et al. Conducted -> ~ conducted. Same thing.
Line 147&149 : Please, merge Fig 5 and Fig 6, and labels them if possible.
Line 194 : specimen a -> specimen A. It seems to be TYPO.
Line 243&264&278 : please merge these figures(Fig. 14~16) and labels them if possible.
Line 296&299 : please merge these two figures(Fig. 17 & 18) and labels them if possible.
Line 321 : Fig.20 -> Fig.19, please check it.
Line 354&355 : please merge these two figures(Fig. 20 & 21) and labels them if possible.
Line 367&374 : please merge these two figures(Fig. 22 & 23) and labels them if possible.
Line 382 ~: same for Figure 24 ~30.
Author Response
We thank the reviewer for his/her constructive criticisms that have helped us to improve our manuscript. The point-by-point response to the comments is given below.
Point 1: There are too many figures. It should be reduced.
Response 1: We have made correction according to the Reviewer’s comments.We have reduced and integrated the pictures as much as possible
Point 2: There is no conclusion. Only summary of this work presented in the conclusion section.
Response 2: We are very sorry for this. We have modified the conclusion to achieve the purpose of more than summary.
Point 3: There is too ordinary explanation for results of numerical simulations. Please, condense and describe only the important facts.
Response 3: We have reduced the analysis of the results and refined the focus.
Point 4: Line 14: what is full name of PSCFST? It seems that CFST means Concrete Filled Steel Tubular. What is PS? Is it prestressing segments? Please check it.
Response 4: PS stands for Precast Segmental. The full name of PSCFST is Precast Segmental Concrete Filled Steel Tubular.
Point 5: Line 124, 125, 132: There is no label for figures. Please insert label for each picture and merge them if possible. Also, No reference for Figure 2. Please check it.3
Response 5: We have made correction according to the Reviewer’s comments and integrated the images and added label under each image.
Point 6: ‘Constitutive relation’ Section, Please provide more information of the model and its parameters.
Response 6: We have added the appropriate model information and its parameters. For example: concrete plastic damage parameters, etc.
Point 7: Line 144 : There is no ‘f’, Is it a typo ?
Response 7: We are very sorry for our incorrect writing. We have made correction according to the Reviewer’s comments.
Point 8: Line 145: 2.0GPa is not a common value for the steel elastic modulus. Please check it
Response 8: We are very sorry for our incorrect writing. It's not 2.0GPa but 2.02*105MPa.
Point 9: What kind of material model and its parameters for concrete material in ABAQUS? Please provide detailed information.
Response 9: We have made correction according to the Reviewer’s comments. We added the expansion angle, flow potential bias, stress ratio, invariant stress ratio and viscosity coefficient.
Point 10: Line 171,173,176 : please merge three figures into one and labels them if possible. Also, use different line styles for the results of left and right side.
Response 10: We have made correction according to the Reviewer’s comments. We have integrated the images and color divided the curves on the left and right.
Point 11: Line 177~184: what is the main fact having different results of numerical simulations for left and right side?
Response 11: There will be some errors during specimen processing, and concrete is a heterogeneous material, which lead to different strain values on the left and right sides.
Point 12: What is the meaning of 103 kN? How did you select this value?
Response 12: Through calculation, we get that the ultimate bearing capacity of CFST pier specimen is 1028kN, so the working condition with initial prestress of 103kN is adopted, that is, the axial compression ratio of initial prestress is 0.1.
Point 13: Line 225: for 0.6 friction coefficient. How did you select this value? Please provide the information how to get this friction coefficient.
Response 13: We selected the friction coefficient by consulting the literature.
Point 14: What is the meaning of 52,103,and 155 kN? How did you select this value?
Response 14: Through calculation, we get that the ultimate bearing capacity of CFST pier specimen is 1028kN, so the working condition with initial prestress of 103kN is adopted, that is, the axial compression ratio of initial prestress is 0.1. When the axial compression ratio of prestress is 0.05, the initial prestress is 52kN, when the axial compression ratio of prestress is 0.15, the initial prestress is 155kN.
Point 15: Line 437 : Conclusion Section, there is no conclusion. Only summary of this work presented in the conclusion section. Please provide the conclusion of your work.
Response 15: We are very sorry for this. We have modified the conclusion to achieve the purpose of more than summary.
Point 16: Line 42 : some Precast Pier -> some precast pier, Is there any meaning on using upper case?
Response 16: We are very sorry for our incorrect writing.Actually,there is no meaning on using upper case.
Point 17: Line 52 : Unboned Prestressed. Same thing.
Response 17: We are very sorry for our incorrect writing.
Point 18: Line 76: Please remove [9]. It was cited doubly.
Response 18: We are very sorry for this.We have removed it.
Point 19: Line 84: ElGawady [12] et al. Conducted -> ~ conducted. Same thing.
Response 19: We are very sorry for our incorrect writing.
Point 20: Line 147&149 : Please, merge Fig 5 and Fig 6, and labels them if possible.
Response 20: We modified Figure 6 and added a formula.
Point 21: Line 194 : specimen a -> specimen A. It seems to be TYPO.
Response 21: We are very sorry for our incorrect writing.
Point 22-23: Line 243&264&278 : please merge these figures(Fig. 14~16) and labels them if possible.
Line 296&299 : please merge these two figures(Fig. 17 & 18) and labels them if possible.
Response 22-23: We have integrated the pictures as much as possible.
Point 24: Line 321 : Fig.20 -> Fig.19, please check it.
Response 24:We are very sorry for our incorrect writing.
Point 25-27: Line 354&355 : please merge these two figures(Fig. 20 & 21) and labels them if possible.
Line 367&374 : please merge these two figures(Fig. 22 & 23) and labels them if possible.
Line 382 ~: same for Figure 24 ~30.
Response 25-27:We have integrated the pictures as much as possible.

Reviewer 2 Report
This research deal with a segmental assembled concrete filled steel tubular pier with external energy dissipation ring (PSCFST) pier against seismic damage for bridge. Based on ABAQUS analysis software, a four segment PSCFST pier model is established, and the pseudo-static comparative analysis is carried out between the traditional PSCFST pier and the PSCFST pier with external energy dissipation ring. At the same time, the seismic performance of pier models with three different control parameters (initial prestress, material strength of energy dissipation ring and section width of energy dissipation ring) under reciprocating loading is analyzed. The results imply a great potential of PSCFST and make clear its fundamental mechanism. However, some parts are difficult to understand because of mainly lack of information of experiment, FE analysis. Please refer to following my comments:
- P4, Fig.2: There is no comment and reference for Fig.2 in the text.
- P5 in section constitutive relation: There are lack of information about boundary condition and friction of the FEM model. For example, how define the friction between steel and concrete?
- P5, line 151: The information of the damage model of the concrete is very limited. Please describe more detail in this method and used parameters.
- P5, line 145-146: Young modules and Poisson’s ratio of the steel is difference from ordinally steel material (Especially in Young modules). Please check again their values in the text. If the values are correct, what kind of steel material is assumed in here?
- P8, line 198 and 200: How do you define the axial compression ration of 0.4 and prestressing force of 103kN?
- P8, line 202: Please describe more fundamental information material Q235. It’s also Q195 and Q345 in line 385 in page 17.
- P8, Fig.10: It’s difficult for me how set the rings in test piece. Please add a describe or figure to understand the connection of rings to the test piece.
- P8, Fig.11: The size of ring is too large comparing with Fig.10. Please check its unit or value.
- P9, Fig.12: How do you define horizontal displacement for the loading control? I guess generally these displacements are related to the yielding displacement of the model.
- P9, line 222: The friction coefficient of 0.6 is surface on segments between concrete and concrete, but how about between concrete and steel in each segment.
- P10, line 242: Did you describe an information of traditional PSCFST? For example, please describe its detail using Fig.10.
- P13, line 305: Although it is limited to between segments got friction contact surface, is there any energy dissipation due to friction, for example, between the steel and concrete in each segment?
- P21, before conclusion: I feel the text is suddenly over without comprehensive Discussions. Do you have any recommendation or summarization to design of your proposed PSCFST based on your consideration for prestressing, material strength, section width.
Author Response
We thank the reviewer for his/her constructive criticisms that have helped us to improve our manuscript. The point-by-point response to the comments is given below.
Point 1: P4, Fig.2: There is no comment and reference for Fig.2 in the text.
Response 1: We have made correction according to the Reviewer’s comments. We have added comments and reference for Fig.2
Point 2: P5 in section constitutive relation: There are lack of information about boundary condition and friction of the FEM model. For example, how define the friction between steel and concrete?
Response 2: We have added the information of boundary conditions in the article. The steel pipe and concrete are bound together by Tie contact.
Point 3: P5, line 151: The information of the damage model of the concrete is very limited. Please describe more detail in this method and used parameters.
Response 3: We have made correction according to the Reviewer’s comments. We added the expansion angle, flow potential bias, stress ratio, invariant stress ratio and viscosity coefficient.
Point 4: P5, line 145-146: Young modules and Poisson’s ratio of the steel is difference from ordinally steel material (Especially in Young modules). Please check again their values in the text. If the values are correct, what kind of steel material is assumed in here?
Response 4: We are very sorry for our incorrect writing. Young modules is 2.02*105MPa and Poisson’s ratio is 0.25.
Point 5: P8, line 198 and 200: How do you define the axial compression ration of 0.4 and prestressing force of 103kN?
Response 5: We select the axial compression ratio through references. Through calculation, we get that the ultimate bearing capacity of CFST pier specimen is 1028kN, so the working condition with initial prestress of 103kN is adopted, that is, the axial compression ratio of initial prestress is 0.1.
Point 6: P8, line 202: Please describe more fundamental information material Q235. It’s also Q195 and Q345 in line 385 in page 17
Response 6: We have added more fundamental information about material Q235, Q195 and Q345.
Point 7: P8, Fig.10: It’s difficult for me how set the rings in test piece. Please add a describe or figure to understand the connection of rings to the test piece.
Response 7: In the actual project, the energy dissipation ring will be connected and fixed with the CFST pier through high-strength bolts and embedded screws in the section. In the finite element method, the energy dissipation ring and the steel pipe wall are contacted with Tie to bring the high-strength bolts.
Point 8: P8, Fig.11: The size of ring is too large comparing with Fig.10. Please check its unit or value.
Response 8: We are very sorry for our incorrect writing, We have modified the size.
Point 9: P9, Fig.12: How do you define horizontal displacement for the loading control? I guess generally these displacements are related to the yielding displacement of the model.
Response 9: The horizontal reciprocating load under constant axial force adopts full displacement control loading, which is actually controlled by displacement angle. Relevant references were attached in this paper.
Point 10: P9, line 222: The friction coefficient of 0.6 is surface on segments between concrete and concrete, but how about between concrete and steel in each segment.
Response 10: We are very sorry for our incorrect writing. Actually ,the steel pipe and concrete are bound together by Tie contact.
Point 11: P10, line 242: Did you describe an information of traditional PSCFST? For example, please describe its detail using Fig.10.
Response 11: We have made correction according to the Reviewer’s comments. For the traditional PSCFST piers, the size structure and prestressing tendon arrangement are the same as the PSCFST piers with external energy dissipation rings, the only difference is that the traditional PSCFST piers do not have external energy dissipation rings.
Point 12: P13, line 305: Although it is limited to between segments got friction contact surface, is there any energy dissipation due to friction, for example, between the steel and concrete in each segment?
Response 12: When ABAQUS simulates 0.4 axial compression ratio to apply vertical force to CFST pier, the value of axial force is 400kN. During the axial compression test of concrete-filled steel tubular column, the axial force was added to 1800kN, and the concrete in the steel tube did not crack or separate from the steel tube wall. Therefore, we use Tie contact in the finite element to simulate the contact between concrete and steel pipe. So there was no friction energy consumption.
Point 13: P21, before conclusion: I feel the text is suddenly over without comprehensive Discussions. Do you have any recommendation or summarization to design of your proposed PSCFST based on your consideration for prestressing, material strength, section width
Response 13: We are very sorry to have made you feel this way. We have included a summary of the results of the parametric analysis in the article.
Special thanks to you for your good comments.

Round 2
Reviewer 1 Report
It seems that the paper has been revised well.